# L-shaped relationship between dietary niacin intake and hearing loss in United States adults: National health and nutrition examination survey

**Zhaocha Gao[1], Yunbing Dai[1], Ting Liu[2], Yungang Wu[1], Xue Zhang[1]\***

**1** Department of Otolaryngology-Head and Neck Surgery, Affiliated Hospital of Jining Medical University, Shandong, China, **2** Department of Endocrinology, Beijing Liangxiang Hospital, Capital Medical University, Beijing, China

\* zhangxue_jyfy@163.com

## Abstract

Hearing loss poses a significant threat to human health, with its prevalence increasing annually. Niacin (vitamin B3) is an essential B vitamin that plays a crucial role in energy metabolism and cellular repair in the body. Additionally, it exerts a protective influence on the cells of the inner ear. A correlation between dietary niacin and hearing loss has been reported; however, the results remain controversial, requiring further investigation. This study aimed to examine the potential association between dietary niacin intake and hearing loss in United States (U.S.) adults, providing a reference for dietary preventive management of hearing loss. In this cross-sectional study, data derived from the National Health and Nutrition Survey for U.S. adults aged 20 to 69 years, spanning the 2011–2012 and 2015–2016 cycles, were used. Logistic regression, restricted cubic spline models, subgroup analyses, and sensitivity analyses were used to assess the stability of the results. A total of 7675 participants were included, of whom 772 (10.1%) exhibited low-frequency hearing loss (LFHL), 1165 (15.2%) had speech-frequency hearing loss (SFHL), and 2816 (36.7%) had high-frequency hearing loss (HFHL). In the final model, the adjusted odds ratios of dietary niacin intake and LFHL, SFHL, and HFHL in Q2 (16.97–23.40 mg/day) were compared with those in Q1 (≤16.96 mg/day) and were 0.73 (0.57–0.92), 0.76 (0.61–0.94), and 0.80 (0.67–0.96), respectively. The relationship between dietary niacin intake and hearing loss (HL) was illustrated via an L-shaped curve in the restricted cubic spline with an inflection point of approximately 23.26 mg/day. The odds ratios for HL in individuals with niacin intake less than 23.26 mg/day were as follows: 0.950 (0.917–0.984) for LFHL, 0.951 (0.921–0.982) for SFHL, and 0.965 (0.939–0.992) for HFHL. To summarize, an "L"-shaped correlation between dietary niacin intake and the occurrence of HL in U.S. adults with an inflection point estimated at approximately 23.26 mg/day was revealed in the present study.

## Introduction

Hearing loss (HL), a prevalent manifestation of ear-related disorders, has emerged as the second most prevalent cause of disability on a global scale [1]. Currently, it is estimated that

**Data availability statement:** "The data used in this study are third-party data from the National Health and Nutrition Examination Survey (NHANES), which is publicly available through the Centers for Disease Control and Prevention (CDC) (https://www.cdc.gov/nchs/nhanes/?CDC_AAref_Val= https://www.cdc.gov/nchs/nhanes/index.htm). The dataset is available from Zenodo at https://zenodo.org/uploads/14633499."

**Funding:** This study was supported by Research Fund project of Academician Helin New Medical Clinical Transformation Workstation(No. JYHL2022FMS11) and Special research plan of attending physician team in Affiliated Hospital of Jining Medical College(No. ZZTD-2022-007).The funders had no role in study design, data collection and analysis, decision to publish, or preparation of the manuscript.

**Competing interests:** The authors have declared that no competing interests exist.

**Abbreviations:** LFHL, low-frequency hearing loss; SFHL, speech-frequency hearing loss; HFHL, high-frequency hearing loss; SGN, spiral ganglion neurons; BDNF, brain-derived neurotrophic factors.

470 million individuals worldwide are affected by HL, with the incidence rate being anticipated to reach 900 million by 2050, primarily due to demographic shifts associated with population aging [2]. In the United States (U.S.), high-frequency hearing loss (HFHL) and speech-frequency hearing loss (SFHL) occur in 31.1% and 14.1% of adults aged 20–69 years, respectively [3]. The prevalence of HL exhibits a twofold increase with each successive decade of age, and it tends to manifest at younger ages [4]. HL has become the primary cause of disability in adults, affecting communication and resulting in mental health problems, increased unemployment, and high medical costs [5–7]. Individuals with HL have higher rates of dementia [8], depression [9], and mortality [10] than the normal population. As a result, exploring the factors affecting HL has become an important area of research, aiming to establish a foundational knowledge base for the early-stage prevention and treatment of hearing impairment.

Modifying dietary intake has been shown to affect the risk of hearing impairment. The primary mechanisms by which diet affects HL are the scavenging of free radicals, maintenance of cochlear microcirculation, and reduction of neuroinflammatory losses [11]. Furthermore, it has been suggested that vitamin B12 and niacin may reduce the risk of HL [12], polyunsaturated fatty acid intake reduces the risk of HL in women [13], and higher dietary intake of niacin, riboflavin, and retinol are associated with a lower prevalence of HL in older adults [14]. However, another study has demonstrated no significant correlation between dietary niacin, folic acid, or vitamin B6 and HL [15] and that the relationship between niacin and HL is a complex topic with inconsistent results from related studies. As a result, using a large sample size from the National Health and Nutrition Survey (NHANES) database, this study aimed to further explore the relationship between dietary niacin and HL. We hypothesize that niacin intake is lower in individuals with HL. Additionally, the dose-response relationship between dietary niacin intake and hearing loss was assessed.

## Methods

### Data sources

The data employed in the present study were obtained from the NHANES, a nationally representative survey conducted by the National Center for Health Statistics. NHANES employed a stratified, multistage probability cluster sampling method to evaluate the health and nutritional status of the mobile population within the U.S. [16]. The NHANES study protocol was approved by the National Center for Health Statistics Research Ethics Review Board, and participants provided written informed consent at enrollment. Ethical approval and consent were not required for secondary analyses. The study adhered to the guidelines for Enhancing the Reporting of Observational Epidemiological Studies.

### Study design and population

NHANES data from the 2011–2012 and 2015–2016 cycles were used in the present cross-sectional study. Only hearing test data for adults aged 20–69 years were available for these two cycles. Consequently, 9,444 participants aged 20–69 years were included in the present study. Of the 9,444, 1,305 with missing hearing test data and 464 with missing dietary niacin data were excluded, and values for covariates with a missingness rate of less than 10% were subjected to multiple interpolations. Finally, 7675 participants with complete hearing tests and dietary niacin intake data were included.

### Hearing loss

Certified audiometric examiners administered auditory tests to participants in a dedicated soundproof room located at a mobile examination center [17]. In the present study, the

outcome variables encompassed LFHL, SFHL, and HFHL. LFHL may cause difficulty in hearing low-pitched speech and environmental sounds. LFHL was delineated as the mean pure-tone audiometry (PTA) exceeding 25 dB HL at 500, 1,000, and 2,000 Hz in both ears, while SFHL may lead to difficulty in hearing and understanding everyday conversations for those with hearing impairments. SFHL was characterized by a PTA exceeding 25 dB HL at 500, 1,000, 2,000, and 4,000 Hz in both ears. HFHL may result in difficulty hearing high-pitched sounds, which can impair speech intelligibility. HFHL was defined as a PTA exceeding 25 dB HL at 3,000, 4,000, 6,000, and 8,000 Hz in both ears [18–21].

## Dietary niacin

The data on dietary niacin intake were collected through 24-hour dietary recall interviews. NHANES dietary surveys were conducted by professionally trained personnel and were based on the Dietary Recall Interview Measurement Guide (https://wwwn.cdc.gov/nchs/data/nhanes/public/2017/manuals/2017_MEC_In-Person_Dietary_Interviewers_Manual.pdf) by converting to U.S. Department of Agriculture standardized reference codes, the food intake aligned with the U.S. Department of Agriculture Food and Nutrient Database for Dietary Studies [22,23]. The nutritional value of each individual's dietary intake was accurately calculated using the Computer-Assisted Dietary Interview system and the Automated Multiple Pass Method. The dietary information provided by the participants was assessed on two occasions by professionals following the Dietary Recall Interview Measurement Guide. The initial assessment was conducted in person at a mobile screening center, while the second assessment was conducted via telephone between three and ten days later. This approach to assessing dietary intake was extensively discussed at a workshop on NHANES data collection procedures and subsequently endorsed by experts in the field [24]. In this study, dietary niacin intake was averaged from two assessments. Participants were then categorized into quartiles based on their dietary niacin intake.

## Other covariates

Potential confounders were identified based on previous studies, clinical experience, p-values less than 0.05 in univariate analyses, and a change of over 10% in the effect sizes of dietary niacin intake on HL before and after adjusting for variables [5,25–29], including age, sex, ethnicity/race, education level, marital status, household income, body mass index (BMI), smoking status, alcohol consumption status, ear infections, tinnitus, hearing protection, noise exposure, hypertension, stroke, diabetes, coronary heart disease, use of dietary supplements, dietary calorie intake, dietary protein intake, dietary carbohydrate intake, and dietary total fat intake. Ethnicity/race was classified as non-Hispanic black, non-Hispanic white, Mexican American, or other. Marital status was classified as married, living alone, or living with a partner. Educational attainment was categorized as < 9 years, 9 to 12 years, or > 12 years. Household income was categorized by a poverty-to-income ratio of ≤ 1.3, 1.3 to 3.5, > 3.5. BMI was calculated using standardized techniques based on weight and height. Smoking status was classified as never smoked, current smoker, or former smoker. Alcohol consumption status was categorized as never drunk or current drinker. In pre-existing conditions (ear infections, tinnitus, hypertension, diabetes, stroke, and coronary heart disease), noise exposure was defined as exposure to loud noises within the past 24 h. Hearing protection was classified according to the frequency of its use during noise exposure. The categories were always, about half the time, seldom, and never. Based on the dietary survey, 24-hour nutritional information was obtained for participants, including total dietary calories, protein, carbohydrates, and total fat. Similar to dietary niacin intake, these variables were

averaged from two assessments to ensure consistency in the evaluation. Dietary supplements were determined through questions regarding the consumption of nutritional supplements and medications in the past month.

## Statistical analysis

Categorical variables are expressed as proportions (%), and continuous variables as mean ± standard deviation (SD). To compare the differences between groups, one-way analysis of variance (normal distribution) and chi-square tests (categorical variables) were performed. Logistic regression models were employed to ascertain the odds ratios (ORs) and corresponding 95% confidence intervals to assess the relationship between dietary niacin intake and hearing impairment. Model 1 was predicated on sex and age. Model 2 was adjusted for sex, age, and comorbidities, including tinnitus, ear infections, hypertension, diabetes, stroke, and coronary heart disease. Model 3 was fully adjusted for age, sex, ethnicity/race, education level, marital status, household income, BMI, smoking status, alcohol consumption status, ear infections, tinnitus, hearing protection, noise exposure, hypertension; stroke, diabetes, coronary heart disease, use of dietary supplements, dietary calorie intake, dietary protein intake, dietary carbohydrate intake, and dietary total fat intake.

The dose-response relationship between dietary niacin intake and hearing impairment was evaluated via restricted cubic spline regression, with a self-help resampling method and likelihood ratio test to determine the inflection point. Additionally, the association threshold between dietary niacin intake and hearing impairment was analyzed using a smoothed, two-stage logistic regression model. Interaction and subgroup analyses were performed using logistic regression models based on subgroups of sex (male vs. female), age (≤45 vs. >45 years), and BMI (<25 vs. 25–30 vs. >30 kg/m²). To assess the robustness of the results, participants with an extreme energy intake that consumed <500 or >5000 kcal per day were excluded. Outliers with a dietary niacin intake outside the mean ± 3 SD (0–25.81 m g/d) range were excluded from the sensitivity analyses.

Because the sample size was determined solely on the data provided, no a priori statistical power estimates were conducted. Data analysis was performed using the statistical software packages R version 4.3.1 (http://www.R-project.org, The R Foundation) and Free Statistics version 1.8 (http://www.clinicalscientists.cn/freestatistics, Beijing, China). Statistical significance was determined via a two-tailed test with a threshold of P < 0.05.

## Results

### Baseline characteristics of the study population

A total of 9,444 participants aged 20–69 years completed the interview. Participants with missing hearing data (n = 1305) and missing niacin dietary intake (n = 464) were excluded. Ultimately, the present cross-sectional study included 7,675 participants from the NHANES database for the 2011–2012 and 2015–2016 cycles. The detailed inclusion and exclusion process can be observed in Fig 1.

The baseline characteristics of excluded and included participants are detailed in the Supplementary Material (S1 Table). The baseline characteristics of all participants according to quartiles of niacin intake are exhibited in Table 1. HL occurred in 37.3% (2863) of the cases. The mean age of the participants was 44.0 ± 14.3 years, and 3809 (49.6%) were male. Those who consumed more niacin tended to be male; younger; married or living with a partner; non-Hispanic white; drank alcohol; never smoked cigarettes; never owned hearing protection; did not experience noise exposure; did not take dietary supplements; had a lower BMI; had a lower prevalence of HL, tinnitus, ear infections, high blood pressure, diabetes, and stroke;

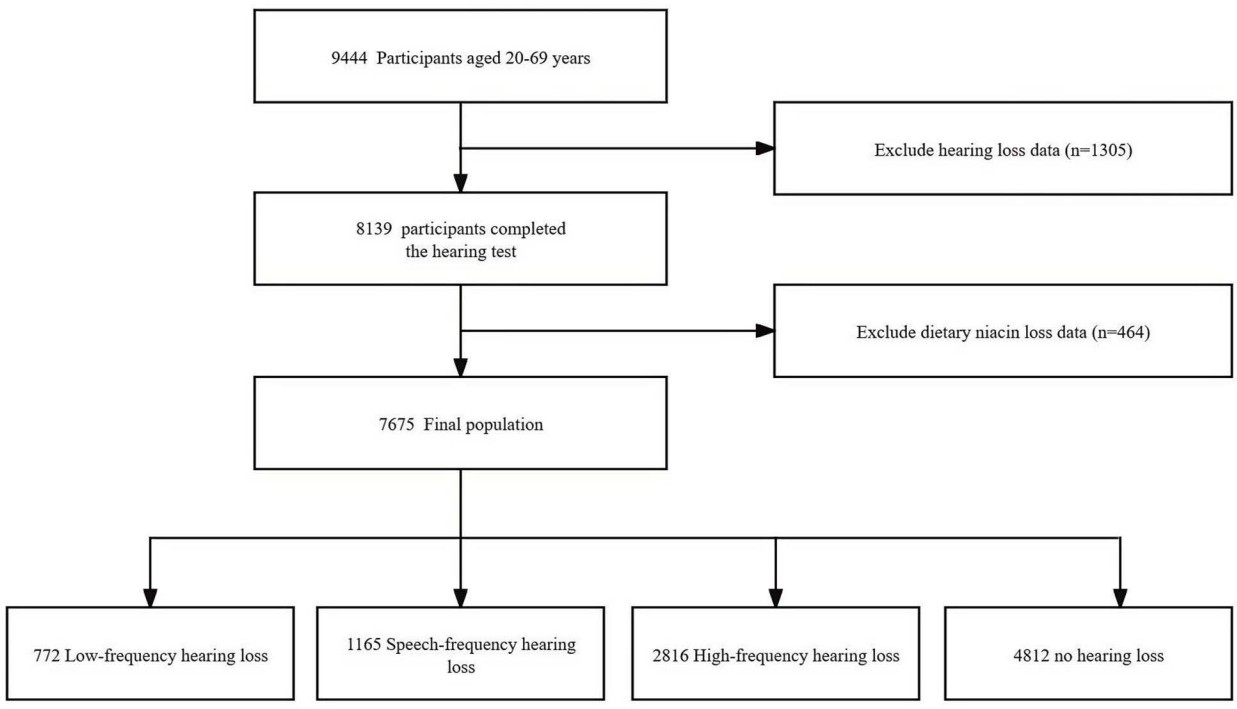

**Fig 1. The study's flow diagram.**

had a higher level of literacy; had a moderate household income; and had a higher intake of calories, protein, carbohydrates, and total fat.

## Relationship between dietary niacin intake and HL

Following univariate analyses, it was revealed that age, sex, race, marital status, household income, BMI, smoking status, drinking status, ear infections, tinnitus, noise exposure, coronary heart disease, stroke, diabetes mellitus, dietary supplementation, energy intake, protein intake, carbohydrate intake, total fat intake, and niacin intake were associated with HL. Marital status was solely linked to HFHL while drinking status was exclusively associated with LFHL. Additionally, ear infections, energy intake, protein intake, carbohydrate intake, and total fat intake were found to be correlated solely with LFHL and SFHL (Table 2).

The model was analyzed using multifactorial logistic regression, with all confounders accounted for in Model 3. The results indicated that dietary niacin intake was negatively associated with the risk of developing HL. Compared with Q1 (<16.69 mg/day), where niacin intake was low, the adjusted ORs of dietary niacin intake and hearing impairment in Q2 (16.97–23.40 mg/day), Q3 (23.41–31.57 mg/day), and Q4 (>31.58 mg/day) were 0.73 (0.57–0.92), 0.83 (0.64–1.09), and 0.81 (0.56–1.15,) for low-frequency hearing impairment, respectively. Speech-frequency hearing impairment for Q2, Q3, and Q4 were 0.76 (0.61–0.94), 0.82 (0.65–1.05), and 0.89 (0.65–1.22), respectively. The adjusted ORs for HFHL were 0.80 (0.67–0.96), 0.94 (0.77–1.16), and 0.89 (0.68–1.15), respectively (Table 3). Consequently, the relationship between dietary niacin intake and HL exhibited an L-shaped curve in the restricted cubic spline, with an inflection point of approximately 23.26 mg/day (Fig 2). In the threshold analyses, the ORs for HL in participants with a niacin intake < 23.26 mg/day were

**Table 1. Baseline characteristics of participants.**

| Variables | Total | Dietary niacin intake,mg/d | | | | p |
|---|---|---|---|---|---|---|
| | | Q1(≤16.96) | Q2(16.97-23.40) | Q3(23.41-31.57) | Q4(>31.58) | |
| NO. | 7675 | 1918 | 1919 | 1919 | 1919 | |
| Sex,n(%) | | | | | | < 0.001 |
| male | 3809 (49.6) | 542 (28.3) | 775 (40.4) | 1055 (55) | 1437 (74.9) | |
| female | 3866 (50.4) | 1376 (71.7) | 1144 (59.6) | 864 (45) | 482 (25.1) | |
| Age,Mean ± SD,years | 44.0 ± 14.3 | 46.1 ± 14.7 | 44.6 ± 14.4 | 44.3 ± 14.1 | 40.9 ± 13.7 | < 0.001 |
| Race/ethnicity,n(%) | | | | | | 0.002 |
| Non-Hispanic white | 2536 (33.0) | 608 (31.7) | 591 (30.8) | 663 (34.5) | 674 (35.1) | |
| Non-Hispanic black | 1907 (24.8) | 522 (27.2) | 510 (26.6) | 458 (23.9) | 417 (21.7) | |
| Mexican American | 1123 (14.6) | 270 (14.1) | 269 (14.0) | 283 (14.7) | 301 (15.7) | |
| Others | 2109 (27.5) | 518 (27.0) | 549 (28.6) | 515 (26.8) | 527 (27.5) | |
| Education level,n(%),years | | | | | | < 0.001 |
| <9 | 605 (7.9) | 191 (10.0) | 161 (8.4) | 138 (7.2) | 115 (6.0) | |
| 9-12 | 948 (12.4) | 297 (15.5) | 209 (10.9) | 214 (11.2) | 228 (11.9) | |
| >12 | 6122 (79.8) | 1430 (74.6) | 1549 (80.7) | 1567 (81.7) | 1576 (82.1) | |
| Marital status,n(%) | | | | | | < 0.001 |
| Married or living with apartner | 4568 (59.5) | 1052 (54.8) | 1164 (60.7) | 1206 (62.8) | 1146 (59.7) | |
| Living alone | 3107 (40.5) | 866 (45.2) | 755 (39.3) | 713 (37.2) | 773 (40.3) | |
| Family income,n(%) | | | | | | < 0.001 |
| Low | 2501 (32.6) | 757 (39.5) | 582 (30.3) | 583 (30.4) | 579 (30.2) | |
| Medium | 2823 (36.8) | 686 (35.8) | 707 (36.8) | 732 (38.1) | 698 (36.4) | |
| High | 2351 (30.6) | 475 (24.8) | 630 (32.8) | 604 (31.5) | 642 (33.5) | |
| BMI,Mean ± SD,kg/m² | 29.5 ± 7.2 | 29.9 ± 7.4 | 29.7 ± 7.4 | 29.4 ± 7.0 | 29.0 ± 6.9 | < 0.001 |
| Smoking status,n(%) | | | | | | < 0.001 |
| Never | 4471 (58.3) | 1153 (60.1) | 1179 (61.4) | 1111 (57.9) | 1028 (53.6) | |
| Current | 1663 (21.7) | 456 (23.8) | 369 (19.2) | 386 (20.1) | 452 (23.6) | |
| Former | 1541 (20.1) | 309 (16.1) | 371 (19.3) | 422 (22) | 439 (22.9) | |
| drink status,n(%) | | | | | | < 0.001 |
| Yes | 5606 (73.0) | 1212 (63.2) | 1340 (69.8) | 1452 (75.7) | 1602 (83.5) | |
| No | 2069 (27.0) | 706 (36.8) | 579 (30.2) | 467 (24.3) | 317 (16.5) | |
| Low-frequency hearing loss, n(%) | | | | | | < 0.001 |
| No | 6903 (89.9) | 1672 (87.2) | 1744 (90.9) | 1726 (89.9) | 1761 (91.8) | |
| Yes | 772 (10.1) | 246 (12.8) | 175 (9.1) | 193 (10.1) | 158 (8.2) | |
| Speech-frequency hearing loss, n (%) | | | | | | 0.003 |
| No | 6510 (84.8) | 1580 (82.4) | 1653 (86.1) | 1625 (84.7) | 1652 (86.1) | |
| Yes | 1165 (15.2) | 338 (17.6) | 266 (13.9) | 294 (15.3) | 267 (13.9) | |
| High-frequency hearing loss, n (%) | | | | | | 0.001 |
| No | 4859 (63.3) | 1171 (61.1) | 1263 (65.8) | 1174 (61.2) | 1251 (65.2) | |
| Yes | 2816 (36.7) | 747 (38.9) | 656 (34.2) | 745 (38.8) | 668 (34.8) | |
| Tinnitus, n (%) | | | | | | 0.592 |
| Yes | 1169 (15.2) | 297 (15.5) | 287 (15) | 307 (16) | 278 (14.5) | |
| No | 6506 (84.8) | 1621 (84.5) | 1632 (85) | 1612 (84) | 1641 (85.5) | |
| Ear infections,n(%) | | | | | | 0.693 |
| Yes | 1872 (24.4) | 470 (24.5) | 478 (24.9) | 449 (23.4) | 475 (24.8) | |
| No | 5803 (75.6) | 1448 (75.5) | 1441 (75.1) | 1470 (76.6) | 1444 (75.2) | |

*(Continued)*

**Table 1.** (Continued)

| Variables | Total | Dietary niacin intake,mg/d | | | | p |
|---|---|---|---|---|---|---|
| | | Q1(≤16.96) | Q2(16.97-23.40) | Q3(23.41-31.57) | Q4(>31.58) | |
| **Diabetes,n(%)** | | | | | | < 0.001 |
| **Yes** | 890 (11.6) | 270 (14.1) | 239 (12.5) | 224 (11.7) | 157 (8.2) | |
| **No** | 6785 (88.4) | 1648 (85.9) | 1680 (87.5) | 1695 (88.3) | 1762 (91.8) | |
| **Hypertension,n(%)** | | | | | | < 0.001 |
| **Yes** | 2387 (31.1) | 672 (35) | 613 (31.9) | 591 (30.8) | 511 (26.6) | |
| **No** | 5288 (68.9) | 1246 (65) | 1306 (68.1) | 1328 (69.2) | 1408 (73.4) | |
| **Coronary heart disease,n(%)** | | | | | | 0.001 |
| **Yes** | 192 (2.5) | 68 (3.5) | 51 (2.7) | 42 (2.2) | 31 (1.6) | |
| **No** | 7483 (97.5) | 1850 (96.5) | 1868 (97.3) | 1877 (97.8) | 1888 (98.4) | |
| **stroke,n(%)** | | | | | | 0.002 |
| **Yes** | 387 (5.0) | 123 (6.4) | 103 (5.4) | 87 (4.5) | 74 (3.9) | |
| **No** | 7288 (95.0) | 1795 (93.6) | 1816 (94.6) | 1832 (95.5) | 1845 (96.1) | |
| **Hearing protection,n(%)** | | | | | | < 0.001 |
| **Always** | 565 (7.4) | 107 (5.6) | 119 (6.2) | 158 (8.2) | 181 (9.4) | |
| **About half the time** | 534 (7.0) | 81 (4.2) | 106 (5.5) | 151 (7.9) | 196 (10.2) | |
| **Seldom** | 452 (5.9) | 69 (3.6) | 99 (5.2) | 127 (6.6) | 157 (8.2) | |
| **Never** | 6124 (79.8) | 1661 (86.6) | 1595 (83.1) | 1483 (77.3) | 1385 (72.2) | |
| **Loud noise exposure in past 24-hour,n(%)** | | | | | | < 0.001 |
| **Yes** | 1121 (14.6) | 209 (10.9) | 244 (12.7) | 261 (13.6) | 407 (21.2) | |
| **No** | 6554 (85.4) | 1709 (89.1) | 1675 (87.3) | 1658 (86.4) | 1512 (78.8) | |
| **Dietary supplements,n(%)** | | | | | | 0.009 |
| **Yes** | 3672 (47.8) | 903 (47.1) | 965 (50.3) | 938 (48.9) | 866 (45.1) | |
| **No** | 4003 (52.2) | 1015 (52.9) | 954 (49.7) | 981 (51.1) | 1053 (54.9) | |
| **Dietary Calorie intake,Mean (SD),kcal/d** | 2098.3 ± 849.3 | 1378.5 ± 472.8 | 1871.4 ± 487.3 | 2229.2 ± 590.6 | 2913.7 ± 907.8 | < 0.001 |
| **Dietary Protein intake,Mean (SD), g/d** | 82.7 ± 36.4 | 48.5 ± 16.0 | 71.1 ± 17.1 | 89.5 ± 20.5 | 121.5 ± 38.2 | < 0.001 |
| **Dietary Carbohydrate intake (g/d),Mean (SD)** | 253.0 ± 108.4 | 178.1 ± 74.8 | 230.5 ± 76.6 | 267.0 ± 87.6 | 336.5 ± 120.9 | < 0.001 |
| **Dietary Fat intake,Mean (SD),g/d** | 79.9 ± 39.8 | 51.2 ± 23.1 | 71.7 ± 26.8 | 85.5 ± 31.9 | 111.1 ± 46.6 | < 0.001 |

0.950 (0.917–0.984) for LFHL, 0.951 (0.921–0.982) for SFHL, and 0.965 (0.939–0.992) for HFHL (Table 4).

## Stratified analyses based on additional variables

Stratified analyses according to sex, age, and BMI to assess the potential impact of the relationship between dietary niacin and hearing impairment resulted in no significant interactions being found in any of the subgroups (Fig 3). Considering multiple tests, the interaction of low-frequency hearing impairment in BMI with a p-value less than 0.05 may not be statistically significant.

## Sensitivity analysis

The exclusion of individuals with a dietary niacin intake > 25.81 m g/d (mean ± 3 SD) resulted in a stable association between dietary niacin intake and HL (S2 Table). The results remained stable after excluding individuals with extreme energy intake (<500 or >5000 kcal/day) (S3 Table).

**Table 2. Association of covariates and hearing loss risk.**

| Variable | Low-frequency hearing loss | | Speech-frequency hearing loss | | High-frequency hearing loss | |
|---|---|---|---|---|---|---|
| | OR_95 CI | P_value | OR_95 CI | P_value | OR_95 CI | P_value |
| **Sex,n(%)** | | | | | | |
| male | 1 (reference) | | 1 (reference) | | 1 (reference) | |
| female | 0.77 (0.66~0.90) | 0.001 | 0.51 (0.45~0.58) | <0.001 | 0.47 (0.42~0.51) | <0.001 |
| **Age,years** | 1.08 (1.07~1.09) | <0.001 | 1.09 (1.09~1.10) | <0.001 | 1.11 (1.10~1.11) | <0.001 |
| **Race/ethnicity,n(%)** | | | | | | |
| Non-Hispanic white | 1 (reference) | | 1 (reference) | | 1 (reference) | |
| Non-Hispanic black | 0.92 (0.75~1.13) | 0.410 | 0.67 (0.56~0.79) | <0.001 | 0.76 (0.67~0.86) | <0.001 |
| Mexican American | 1.28 (1.03~1.60) | 0.028 | 1.10 (0.92~1.33) | 0.296 | 1.08 (0.93~1.24) | 0.317 |
| Others | 1.01 (0.83~1.23) | 0.910 | 0.84 (0.71~0.98) | 0.030 | 0.79 (0.70~0.89) | <0.001 |
| **Education level,n(%),years** | | | | | | |
| <9 | 1 (reference) | | 1 (reference) | | 1 (reference) | |
| 9-12 | 0.75 (0.57~0.98) | 0.038 | 0.72 (0.57~0.91) | 0.006 | 0.66 (0.54~0.81) | <0.001 |
| >12 | 0.42 (0.34~0.53) | <0.001 | 0.38 (0.31~0.46) | <0.001 | 0.37 (0.31~0.43) | <0.001 |
| **Marital status,n(%)** | | | | | | |
| Married or living with a partner | 1 (reference) | | 1 (reference) | | 1 (reference) | |
| Living alone | 1.00 (0.86~1.16) | 0.968 | 0.90 (0.79~1.02) | 0.111 | 0.83 (0.75~0.91) | <0.001 |
| **Family income,n(%)** | | | | | | |
| Low | 1 (reference) | | 1 (reference) | | 1 (reference) | |
| Medium | 0.78 (0.66~0.93) | 0.007 | 0.81 (0.70~0.94) | 0.005 | 0.85 (0.76~0.94) | 0.003 |
| High | 0.73 (0.61~0.88) | 0.001 | 0.73 (0.62~0.85) | <0.001 | 0.78 (0.69~0.87) | <0.001 |
| **BMI,kg/m²** | 1.02 (1.01~1.03) | <0.001 | 1.02 (1.01~1.02) | <0.001 | 1.02 (1.01~1.02) | <0.001 |
| **Smoking status,n (%)** | | | | | | |
| Never | 1 (reference) | | 1 (reference) | | 1 (reference) | |
| Current | 1.36 (1.13~1.64) | 0.001 | 1.58 (1.35~1.85) | <0.001 | 1.69 (1.50~1.89) | <0.001 |
| Former | 1.79 (1.50~2.14) | <0.001 | 2.29 (1.97~2.65) | <0.001 | 2.49 (2.22~2.81) | <0.001 |
| **drink status,n(%)** | | | | | | |
| Yes | 1 (reference) | | 1 (reference) | | 1 (reference) | |
| No | 1.18 (1.01~1.39) | 0.041 | 0.92 (0.80~1.07) | 0.280 | 1.03 (0.92~1.14) | 0.636 |
| **Tinnitus,n(%)** | | | | | | |
| Yes | 1 (reference) | | 1 (reference) | | 1 (reference) | |
| No | 0.27 (0.23~0.32) | <0.001 | 0.28 (0.24~0.32) | <0.001 | 0.32 (0.28~0.36) | <0.001 |
| **Ear infections,n(%)** | | | | | | |
| Yes | 1 (reference) | | 1 (reference) | | 1 (reference) | |
| No | 0.63 (0.54~0.74) | <0.001 | 0.75 (0.66~0.87) | <0.001 | 0.92 (0.82~1.02) | 0.121 |
| **Diabetes,n(%)** | | | | | | |
| Yes | 1 (reference) | | 1 (reference) | | 1 (reference) | |
| No | 0.37 (0.31~0.45) | <0.001 | 0.31 (0.27~0.37) | <0.001 | 0.27 (0.23~0.31) | <0.001 |
| **Hypertension,n(%)** | | | | | | |
| Yes | | | | | | |
| No | 0.44 (0.37~0.51) | <0.001 | 0.41 (0.36~0.46) | <0.001 | 0.33 (0.30~0.37) | <0.001 |
| **stroke,n(%)** | | | | | | |
| Yes | 1 (reference) | | 1 (reference) | | 1 (reference) | |
| No | 0.35 (0.25~0.50) | <0.001 | 0.27 (0.20~0.37) | <0.001 | 0.23 (0.17~0.31) | <0.001 |
| **Coronary heart disease,n(%)** | | | | | | |
| Yes | 1 (reference) | | 1 (reference) | | 1 (reference) | |
| No | 0.35 (0.27~0.45) | <0.001 | 0.29 (0.23~0.36) | <0.001 | 0.23 (0.19~0.29) | <0.001 |

*(Continued)*

**Table 2.** (Continued)

| Variable | Low-frequency hearing loss | | Speech-frequency hearing loss | | High-frequency hearing loss | |
|---|---|---|---|---|---|---|
| | OR_95 CI | P_value | OR_95 CI | P_value | OR_95 CI | P_value |
| **Hearing protection,n(%)** | | | | | | |
| **Always** | 1 (reference) | | 1 (reference) | | 1 (reference) | |
| **About half the time** | 0.80 (0.53~1.21) | 0.297 | 1.01 (0.73~1.40) | 0.937 | 0.96 (0.75~1.22) | 0.722 |
| **Seldom** | 0.90 (0.59~1.36) | 0.605 | 1.00 (0.71~1.40) | 0.999 | 0.83 (0.64~1.07) | 0.153 |
| **Never** | 1.00(0.75~1.32) | 0.984 | 0.93 (0.73~1.18) | 0.544 | 0.85 (0.72~1.02) | 0.078 |
| **Loud noise exposure in past 24-hour,n(%)** | | | | | | |
| **Yes** | 1 (reference) | | 1 (reference) | | 1 (reference) | |
| **No** | 1.54 (1.21~1.95) | <0.001 | 1.69 (1.37~2.07) | <0.001 | 2.08 (1.80~2.41) | <0.001 |
| **Dietary supplements taken, n(%)** | | | | | | |
| **Yes** | 1 (reference) | | 1 (reference) | | 1 (reference) | |
| **No** | 0.85 (0.73~0.99) | 0.036 | 0.84 (0.75~0.96) | 0.008 | 0.85 (0.77~0.93) | <0.001 |
| **Dietary calorie intake,kcal/d** | 1.00 (1.00~1.00) | 0.001 | 1.00 (1.00~1.00) | 0.042 | 1.00 (1.00~1.00) | 0.278 |
| **Dietary protein intake,g/d** | 1.00(0.99~1.00) | <0.001 | 1.00 (1.00~1.00) | 0.028 | 1.00 (1.00~1.00) | 0.277 |
| **Dietary carbohydrate intake,g/d** | 1.00 (1.00~1.00) | 0.001 | 1.00 (1.00~1.00) | 0.009 | 1.00 (1.00~1.00) | 0.171 |
| **Dietary total fat intake,g/d** | 1.00 (1.00~1.00) | 0.010 | 1.00 (1.00~1.00) | 0.246 | 1.00 (1.00~1.00) | 0.300 |
| **Dietary niacin intake,mg/day** | 0.99 (0.98~0.99) | <0.001 | 0.99 (0.99~1.00) | 0.004 | 0.99 (0.99~1.00) | 0.005 |

**Table 3. Association of dietary niacin intake with hearing loss.**

| Dietary niacin intake,mg/day | NO. | crude | P_value | Model 1 | P_value | Model 2 | P_value | Model 3 | P_value |
|---|---|---|---|---|---|---|---|---|---|
| **Low-frequency hearing loss** | | | | | | | | | |
| **Quartiles** | | | | | | | | | |
| **Q1(≤16.96)** | 1918 | 1(Ref) | | 1(Ref) | | 1(Ref) | | 1(Ref) | |
| **Q2(16.97-23.40)** | 1919 | 0.68 (0.56~0.84) | <0.001 | 0.71 (0.57~0.88) | 0.002 | 0.68 (0.54~0.85) | 0.001 | 0.73 (0.57~0.92) | 0.009 |
| **Q3(23.41-31.57)** | 1919 | 0.76 (0.62~0.93) | 0.007 | 0.79 (0.63~0.98) | 0.029 | 0.76 (0.61~0.95) | 0.015 | 0.83 (0.64~1.09) | 0.185 |
| **Q4(>31.58)** | 1919 | 0.61 (0.49~0.75) | <0.001 | 0.75 (0.59~0.95) | 0.017 | 0.74 (0.58~0.95) | 0.016 | 0.81 (0.56~1.15) | 0.233 |
| **Speech-frequency hearing loss** | | | | | | | | | |
| **Quartiles** | | | | | | | | | |
| **Q1(≤16.96)** | 1918 | 1(Ref) | | 1(Ref) | | 1(Ref) | | 1(Ref) | |
| **Q2(16.97-23.40)** | 1919 | 0.75 (0.63~0.90) | 0.001 | 0.73 (0.60~0.89) | 0.002 | 0.71(0.58~0.86) | 0.001 | 0.76 (0.61~0.94) | 0.012 |
| **Q3(23.41-31.57)** | 1919 | 0.85 (0.71~1.00) | 0.055 | 0.78 (0.64~0.94) | 0.011 | 0.76 (0.62~0.92) | 0.006 | 0.82 (0.65~1.05) | 0.110 |
| **Q4(>31.58)** | 1919 | 0.76 (0.63~0.90) | 0.002 | 0.81 (0.66~0.99) | 0.043 | 0.81 (0.66~1.01) | 0.058 | 0.89 (0.65~1.22) | 0.466 |
| **High-frequency hearing loss** | | | | | | | | | |
| **Quartiles** | | | | | | | | | |
| **Q1(≤16.96)** | 1918 | 1(Ref) | | 1(Ref) | | 1(Ref) | | 1(Ref) | |
| **Q2(16.97-23.40)** | 1919 | 0.81 (0.71~0.93) | 0.002 | 0.75 (0.64~0.89) | 0.001 | 0.75 (0.63~0.89) | 0.001 | 0.80 (0.67~0.96) | 0.018 |
| **Q3(23.41-31.57)** | 1919 | 0.99 (0.87~1.13) | 0.937 | 0.89 (0.76~1.06) | 0.193 | 0.90 (0.76~1.06) | 0.209 | 0.94 (0.77~1.16) | 0.579 |
| **Q4(31.548)** | 1919 | 0.84 (0.73~0.95) | 0.008 | 0.84 (0.71~1.01) | 0.061 | 0.86 (0.71~1.02) | 0.087 | 0.89 (0.68~1.15) | 0.354 |

Model 1 adjusted for age, sex.

Model 2 adjusted for age, sex, tinnitus, ear infections, hypertension, diabetes, stroke, coronary heart disease.

Model 3 adjusted for age, sex, tinnitus, ear infections, hypertension, diabetes, stroke, coronary heart disease, race/ethnicity, education level, household income, marital status, body mass index, smoking status, drink status, noise exposure, hearing protection, dietary energy intake, dietary protein intake, dietary carbohydrate intake, dietary total fat intake, dietary supplements.

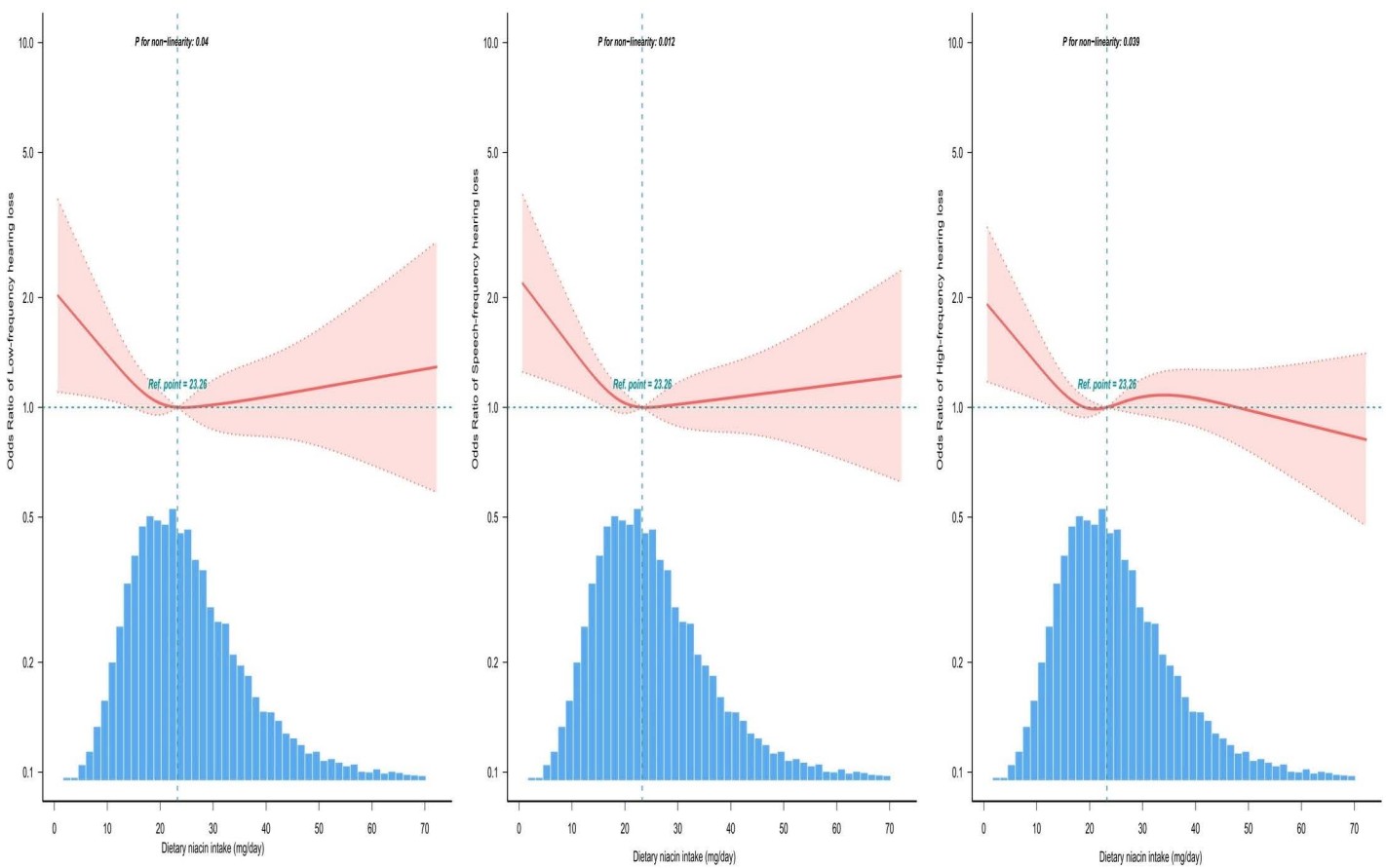

**Fig 2. Restricted cubic spline model for the ratio of dietary niacin intake to hearing loss.** Adjusted for age, sex, tinnitus, ear infections, hypertension, diabetes, stroke, coronary heart disease, race/ethnicity, education level, household income, marital status, body mass index, smoking status, drink status, noise exposure, hearing protection, dietary energy intake, dietary protein intake, dietary carbohydrate intake, dietary fat intake, dietary supplements. Solid and dashed lines represent the predicted value and 95%confidence intervals.

**Table 4. Threshold effect analysis of the relationship of niacin intake with hearing loss.**

| Dietary niacin intake,mg/day | Adjusted Model | |
|---|---|---|
| | OR (95% CI) | p-value |
| **Low-frequency hearing loss** | | |
| Niacin < 23.26 | 0.950 (0.917 ~ 0.984) | 0.0043 |
| Niacin ≥ 23.26 | 1.005 (0.992 ~ 1.019) | 0.4213 |
| **Likelihood Ratio test** | – | 0.0020 |
| **Speech-frequency hearing loss** | | |
| Niacin < 23.26 | 0.951 (0.921 ~ 0.982) | 0.0021 |
| Niacin ≥ 23.26 | 1.006 (0.995 ~ 1.017) | 0.3025 |
| **Likelihood Ratio test** | – | 0.0010 |
| **High-frequency hearing loss** | | |
| Niacin < 23.26 | 0.965 (0.939 ~ 0.992) | 0.0102 |
| Niacin ≥ 23.26 | 0.996(0.987 ~ 1.005) | 0.3666 |
| **Likelihood Ratio test** | – | 0.0040 |

OR, odds ratio; CI, confidence interval. Adjusted for age, sex, tinnitus, ear infections, hypertension, diabetes, stroke, coronary heart disease, race/ethnicity, education level, household income, marital status, body mass index, smoking status, drink status, noise exposure, hearing protection, dietary energy intake, dietary protein intake, dietary carbohydrate intake, dietary total fat intake, dietary supplements. Only 99% of the data is displayed.

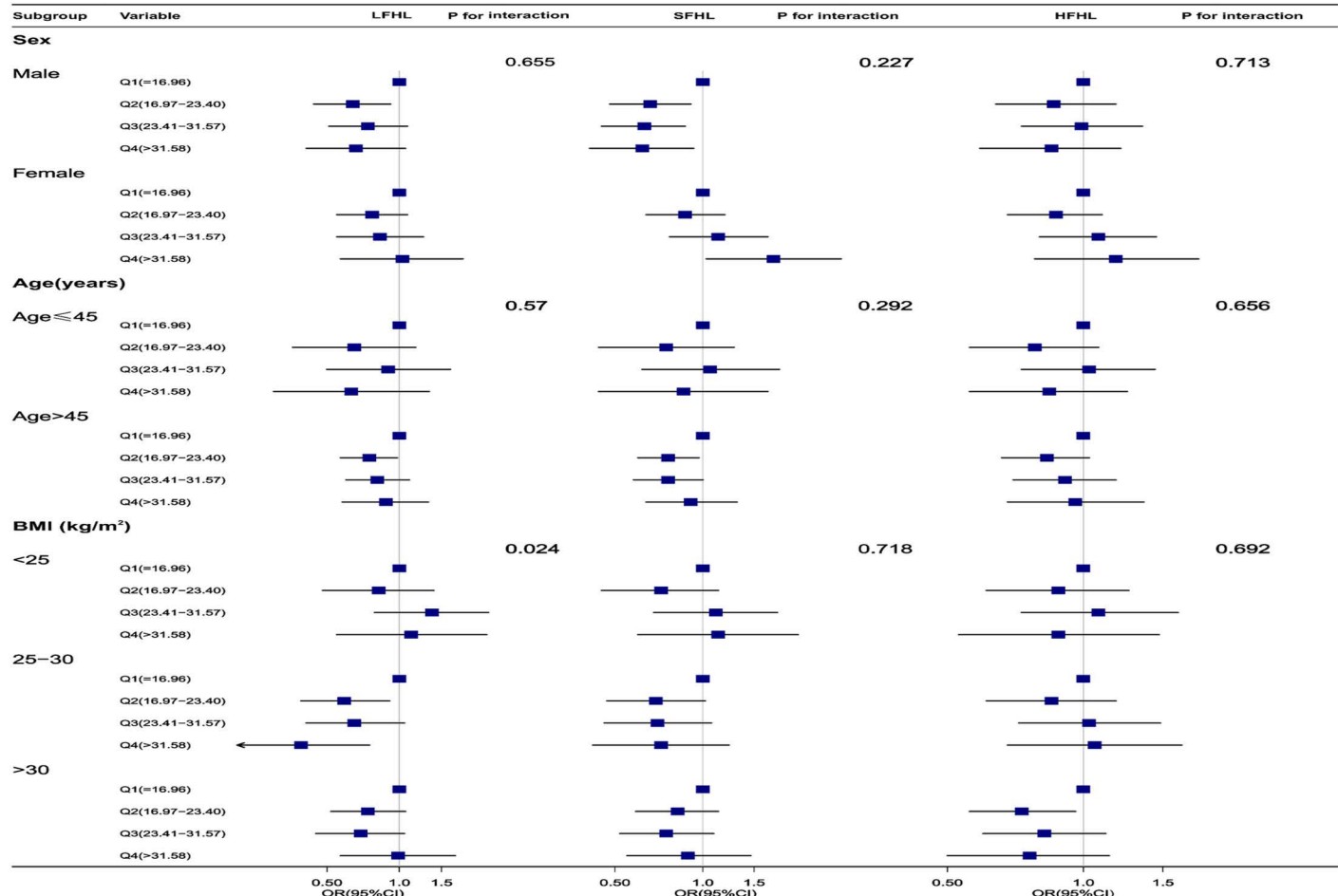

**Fig 3. Association between niacin intake with hearing loss.** Except for the stratification component itself,each stratification factor was adjusted for all other variables, including age, sex, tinnitus, ear infections, hypertension, diabetes, stroke, coronary heart disease, race/ethnicity, education level, household income, marital status, body mass index, smoking status, drink status, noise exposure, hearing protection, dietary energy intake, dietary protein intake, dietary carbohydrate intake, dietary total fat intake, and dietary supplements. Low-frequency hearing loss(LFHL),Speech-frequency hearing loss(SFHL),High-frequency hearing loss(HF-HL),Squares represent odds ratios (ORs), with horizontal lines indicating 95% confidence intervals (CIs).

## Discussion

The findings of this study indicated a negative correlation between dietary niacin intake and HL.To ensure the reliability of the results, potential confounding variables were controlled for in Model 3 of our analyses. The reliability and stability of the results were evaluated using multifactorial logistic regression, stratification, and sensitivity analyses. Furthermore, our study identified an L-shaped relationship between dietary niacin intake and HL, with an inflection point at approximately 23.26 mg/day. When niacin intake was < 23.26 mg/d, the risk of HL decreased with each 1-mg increase in niacin intake (LFHL 0.05 percent, SFHL 0.049 percent, HFHL 0.035 percent). Furthermore, when niacin intake was ≥ 23.26 mg/d, the propensity for HL no longer exhibited a diminishing trend with higher niacin intake.

The L-shaped relationship between niacin intake and hearing protection suggests that moderate consumption may be beneficial, whereas excessive intake might not offer additional advantages. This finding is generally consistent with, but higher than, the recommended daily intake levels set forth by the U.S. Dietary Guidelines 2020–2025 (16 mg/day for men and

14 mg/day for women) [30]. This suggests that exceeding these recommended daily intake levels may confer additional health benefits. Nevertheless, an intake of 23.26 mg/day remains below the upper tolerable limit of 35 mg/day for dietary niacin, indicating that this level of intake is safe. This finding supports the possibility that niacin intake can be slightly higher than the current dietary guideline recommendation. However, further studies are needed to confirm its long-term effects and safety.

The existing literature on the effects of niacin on HL is limited, and the results are inconclusive. The impact of different frequencies of HL on individuals is well documented. LFHL primarily affects the perception of low-pitched sounds and is usually associated with middle ear problems. In contrast, SFHL affects sound perception in the speech-frequency range, resulting in direct impairment of daily communication and speech comprehension. Finally, HFHL is typically caused by age-related HL or noise exposure and affects the perception of high-pitched sounds [19–21,31]. In the present study, after adjusting for potential confounders, niacin intake was observed to be significantly and negatively associated with different frequencies of HL. Nakagawa-Nagahama et al. identified a negative association between blood niacin levels and HL in their analysis of a recent study conducted among 42 older Japanese men [32]. In a cross-sectional study in Malaysia based on 253 participants aged 60 years and older, Ooi et al. evidenced that reduced niacin intake increased the risk of HL [33]. In line with the present findings, Kim et al. have found that dietary niacin intake was associated with a reduced risk of HL in a Korean population of older adults > 65 years [14]. However, in another cross-sectional study on a population aged 50–80 years in Korea, niacin was found to be negatively associated with HL following univariate analysis, and dietary niacin intake was not significantly associated with HL after adjusting for confounders [34]. The differences in findings can be explained by variations in dietary intake assessment techniques, methods of reporting examinations for HL, and study populations. The association between niacin and HL in middle-aged and older populations was reported in all of the aforementioned studies. The present study leverages the substantial sample size of NHANES to conduct an in-depth investigation into the potential association between niacin and HL within the 20–69 age group while also examining the dose-response relationship between these variables. This research aims to bridge gaps in previous studies in this area.

HL is caused by the loss of both vascular volume and spiral ganglion neurons (SGN) due to loss or damage of cells around the inner ear, and loss of SGN is usually associated with various types of HL [35, 36]. Neurotrophic factor support ensures SGN survival and reduces SGN degeneration, especially for brain-derived neurotrophic factors (BDNF) [36, 37]. In a model predicated on strokes in rats, niacin treatment increased the effects of neuronal synapse growth and BDNF/proto-myosin-related kinase B expression [38]. Additionally, niacin intake may reduce the risk of HL by upregulating BDNF, and when dietary niacin intake is high, vascular endothelial cell function increases to ensure blood flow supply to the inner ear and to avoid degeneration from SGN damage [39], suggesting a potential biological mechanism for increasing niacin intake to prevent HL. However, additional prospective studies are required to validate the preventive impact of dietary niacin intake on the occurrence of HL.

It has been suggested that a nutritious diet can help reduce the incidence of HL, and Sharon et al. evidenced in a large cohort study based on women that adherence to a Mediterranean dietary pattern reduced the risk of HL [40]. The Mediterranean diet is typically rich in niacin and includes a large amount of plant foods such as legumes, nuts, vegetables, and cereals, as well as seafood and fish, lamb, and dairy products [41, 42]. Nevertheless, the American diet, commonly called the Western diet, is distinguished by its high prevalence of animal proteins and refined carbohydrates, among other components [43]. As a result of the present study, the hypothesis that dietary pattern modifications aimed at enhancing niacin

intake could serve as a promising intervention strategy to mitigate the risk of HL in the adult population of the U.S. was made.

There are several strengths to the present study. First, a large and nationally representative sample size was used. Second, adjustments were made for known and potential confounders where possible. Moreover, the dose-response relationship between dietary niacin intake and HL was discussed. Despite the strengths, there are also various limitations to the present study. First, based on cross-sectional studies, a definitive causal relationship between niacin and HL could not be ascertained. Second, the potential for residual confounding effects cannot be entirely discounted, even after accounting for a larger number of potential confounders. Third, dietary intake acquisition by recall may be subject to memory bias. However, the probability of such bias is low as the survey obtained detailed information regarding food type and quantity and was superior to food frequency surveys [44, 45]. Given these limitations, well-designed prospective studies are needed to validate the findings further.

## Conclusions

In summary, an L-shaped relationship between dietary niacin intake and the risk of developing HL in U.S. adults with an inflection point of approximately 23.26 mg/d was observed. These findings may hold significant implications for research focused on influencing HL through dietary adjustments.

## Supporting information

**S1 Table. Baseline characteristics of included and excluded participants.**
(DOC)

**S2 Table. Association of dietary niacin intake with hearing loss. Model 1 adjusted for age, sex.**
(DOC)

**S3 Table. Association of dietary niacin intake with hearing loss.** Model 1 adjusted for age, sex. Model 2 adjusted for age, sex, tinnitus, ear infections, hypertension, diabetes, stroke, coronary heart disease. Model 3 adjusted for age, sex, tinnitus, ear infections, hypertension, diabetes, stroke, coronary heart disease, race/ethnicity, education level, household income, marital status, body mass index, smoking status, drink status, noise exposure, hearing protection, dietary energy intake, dietary protein intake, dietary carbohydrate intake, dietary total fat intake, dietary supplements.
(DOC)

## Acknowledgements

We gratefully thank Xiaolan Zhang, Department of Pediatrics, Third People's Hospital of Liaocheng, Liaocheng, Shandong Province, China.Jinbao Ma, Department of Drug-Resistant Tuberculosis, Xi'an Chest Hospital, China, and Jie Liu, Department of Vascular and Endovascular Surgery, General Hospital of the Chinese People's Liberation Army, China, for contributing to the statistical support, study design consultation, and comments on the manuscript.

## Author contributions

**Conceptualization:** Zhaocha Gao, Yunbing Dai, Yungang Wu, Xue Zhang.

**Data curation:** Ting Liu, Xue Zhang.

**Funding acquisition:** Yungang Wu, Xue Zhang.

**Investigation:** Yunbing Dai, Xue Zhang.

**Writing – original draft:** Zhaocha Gao, Yunbing Dai, Yungang Wu, Xue Zhang.

**Writing – review & editing:** Zhaocha Gao, Yunbing Dai, Yungang Wu, Xue Zhang.

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
