## [Decision Letter · Decision Letter 0]

23 Jul 2024

PONE-D-24-05737L-shaped relationship between dietary niacin intake and hearing loss in US adults: National Health and Nutrition Examination Survey

PLOS ONE

Dear Dr. zhang,

Thank you for submitting your manuscript to PLOS ONE. After careful consideration, we feel that it has merit but does not fully meet PLOS ONE’s publication criteria as it currently stands. Therefore, we invite you to submit a revised version of the manuscript that addresses the points raised during the review process.

Dear Authors 

you should list the potential confounders that were accounted and describe how the potential confounders were accounted for? Authors are also requested to provide details how dietary niacin was obtained? What was the goal of assessing meal frequency? Please explain how niacin will be assessed among other vitamins/micronutrients is not yet clear in background section. In method Section, could you provide more details on how noise exposure and hearing protection were determined?

Also carefully do discussion according to results obtained in the current study

We look forward to receiving your revised manuscript.

Kind regards,

Aziz ur Rahman Muhammad

Academic Editor

PLOS ONE

Additional Editor Comments:

The author should list the potential confounders that were accounted and describe how the potential confounders were accounted for? Authors are also requested to provide details how dietary niacin was obtained? What was the goal of assessing meal frequency? Please explain how niacin will be assessed among other vitamins/micronutrients is not yet clear in background section. In method Section, could you provide more details on how noise exposure and hearing protection were determined?

Also carefully do discussion according to results obtained in the current study

Reviewers' comments:

Reviewer's Responses to Questions

**Comments to the Author**

1. Is the manuscript technically sound, and do the data support the conclusions?

Reviewer #1: Yes

Reviewer #2: Yes

2. Has the statistical analysis been performed appropriately and rigorously? 

Reviewer #1: Yes

Reviewer #2: I Don't Know

3. Have the authors made all data underlying the findings in their manuscript fully available?

Reviewer #1: Yes

Reviewer #2: Yes

4. Is the manuscript presented in an intelligible fashion and written in standard English?

Reviewer #1: Yes

Reviewer #2: Yes

5. Review Comments to the Author

Reviewer #1: (This is an interesting manuscript on the “L-shaped relationship between dietary niacin intake and hearing loss in US adults: National Health and Nutrition Examination Survey." The manuscript is well-written showing empirical evidence and meeting all standards of quantitative research. However, in the result section, the author stated that after accounting for all potential confounders there was a significant negative association between dietary niacin intake and HL but failed to indicate those confounders and how they were accounted for.

My recommendation

1. The author should list the potential confounders that were accounted for.

2. The author should describe how the potential confounders were accounted for.)

Reviewer #2: Method Section:

It is still unclear how dietary niacin was obtained. Could you please provide more details?

What is the goal of assessing meal frequency (line 113)? Additionally, the term "four group" in line 114 could be changed to "categorized into quartiles" for better clarity.

Background Section:

The explanation of how niacin will be assessed among other vitamins/micronutrients is not yet clear. Please elaborate on the importance of niacin and why it is a focus of this study.

Method Section - Noise Exposure and Hearing Protection:

Could you provide more details on how noise exposure and hearing protection were determined?

Method and Discussion Sections - Types of Hearing Loss:

Could you briefly explain the differences between low-frequency hearing loss (LFHL), speech-frequency hearing loss (SFHL), and high-frequency hearing loss (HFHL) in both the method and discussion sections?

Logistic Regression and Covariates:

How were the covariates included in the logistic regression determined? Was it based on a p-value threshold (e.g., p<0.2) or another justification?

Assessment of Other Vitamins:

Why were other vitamins, such as B12 and folate, not assessed in the study?

Discussion Section:

Is the inflection point of 23.26 mg/day of niacin intake comparable to current dietary guidelines for the adult population? If so, please add this information to the discussion.

What is the goal of determining an L-shaped association? Please explain this in the discussion section.

6. PLOS authors have the option to publish the peer review history of their article (what does this mean? ). If published, this will include your full peer review and any attached files.

**Do you want your identity to be public for this peer review?** For information about this choice, including consent withdrawal, please see our Privacy Policy .

Reviewer #1: **Yes: ** Felix Gumaayiri Aabebe

Reviewer #2: No

---

## [Author Response · Author response to Decision Letter 1]

21 Aug 2024

Re:  PONE-D-24-05737

Dear Dr. Aziz ur Rahman Muhammad and Reviewers,

We, the co-authors, would like to express our sincere gratitude for your thorough review and valuable comments. We are confident that the revisions made in response to your suggestions have strengthened our manuscript, entitled "L-shaped relationship between dietary niacin intake and hearing loss in US adults: National Health and Nutrition Examination Survey" (ID: PONE-D-24-05737), which we are resubmitting for consideration in PLOS ONE. We are hopeful for a favorable review of the revised submission.

Enclosed, please find a document providing detailed responses to each comment. We have made the corresponding revisions to our manuscript without altering the statistical analyses or the results. We have highlighted the revised sections in red for ease of review. We look forward to your positive response to the revised paper submitted here.

Your time and attention to our research are greatly appreciated, and we stand ready to provide any further information or clarification that may be needed during the review process. It is our earnest hope that the publication of this study in your prestigious journal will make a significant contribution to the academic community.

We declare that there are no conflicts of interest concerning this work, and all authors have read and endorsed the revised manuscript for submission to PLOS ONE. Please feel free to reach out to us for any additional assistance.

Thank you and best regards.

Yours sincerely,

Xue zhang

Email: zhangxue_jyfy@163.com

Mailing address: Affiliated Hospital of Jining Medical University, 89 Guhuai Road, Rencheng District, Jining, Shandong Province, China;

Reviewer: 1

Dear Reviewer,

Thank you very much for your time involved in reviewing the manuscript and your positive comments and valuable suggestions to improve the quality of our manuscript. We have proofread the manuscript carefully. And we hope the revised manuscript could be acceptable for you.

COMMENTS FOR THE AUTHOR:

This is an interesting manuscript on the “L-shaped relationship between dietary niacin intake and hearing loss in US adults: National Health and Nutrition Examination Survey." The manuscript is well-written showing empirical evidence and meeting all standards of quantitative research. However, in the result section, the author stated that after accounting for all potential confounders there was a significant negative association between dietary niacin intake and HL but failed to indicate those confounders and how they were accounted for.My recommendation

1.The author should list the potential confounders that were accounted for.

Response 1）

We would like to express our gratitude for your comprehensive and detailed review, which provided positive affirmation and high-quality guidance. We concur with your suggestions and have implemented substantial revisions to the manuscript. For your reference, we have amended this in the manuscript and highlighted it in red. Please refer to lines123-130 on page 7-8 ,lines197-199 on page 11 and lines 222-226 on page 12 of the revised manuscript. We look forward to your favourable consideration response.

In our study, we considered a comprehensive range of potential confounding variables to ensure the robustness and validity of our findings. The adjusted variables include:

Demographic Factors:age, sex. race/ethnicity,

Socioeconomic Status:education level,marital status,household income.

Health and Lifestyle Factors:body mass index,drink status,smoking status.

Dietary Factors:dietary energy intake, dietary protein intake, dietary carbohydrate intake, dietary fat intake, dietary supplements

Medical History:tinnitus, ear infection, hypertension, diabetes, stroke, coronary heart disease , noise exposure, hearing protection.

To account for these potential confounding variables, we employed multifactor logistic regression analysis. Each confounding variable was incorporated into the model as a covariate to adjust for its effect on the primary outcome. The effects of these potential confounding variables were controlled in this manner. Additionally, we conducted stratification and sensitivity analyses to further ensure the robustness of our findings.

2.The author should describe how the potential confounders were accounted for.

Response 2）

We are grateful for your valuable input. In our study, we took great care to identify and control for numerous potential confounding variables, thereby ensuring the accuracy and reliability of our results. The specific measures we employed are outlined below:

①Potential confounders were identified based on previous studies, clinical experience, p-values less than 0.05 in univariate analyses, and a change of more than 10% in the effect sizes of dietary niacin intake on hearing loss before and after adjusting for variables[1-6],including age; sex; ethnicity/race; education level; marital status; household income; body mass index (BMI); smoking status; alcohol consumption status; ear infections; tinnitus; hearing protection; noise exposure; hypertension; stroke; diabetes; coronary heart disease; use of dietary supplements; dietary calorie intake; dietary protein intake;dietary carbohydrate intake;dietary fat intake.

②Our study utilizes data from the National Health and Nutrition Examination Survey (NHANES), which provides comprehensive coverage, high-quality data on demographics and health behaviors, laboratory and physical examination data, and a nationally representative sample. This enables us to control for confounding variables with greater precision and comprehensiveness, thereby contributing to more reliable and accurate study conclusions.

③During the data analysis phase, we employed multifactor logistic regression analysis to incorporate the aforementioned confounders as control variables into the model, thereby minimizing their impact on the primary results. Additionally, we conducted subgroup analyses and multiple sensitivity analyses to further control for the effects of potential confounders, ensuring the accuracy and reliability of the results.

④Our methodology is informed by the studies of Huanxian Liu et al. (2022), Yi Zhang et al. (2023), and Yafei Mao et al. (2022), who also employed multifactor logistic regression analyses, subgroup analyses, and sensitivity analyses to control for confounding factors in similar research[6-8].

The reliability of our findings was ensured through the implementation of a rigorous study design and robust data analysis methods that effectively controlled for potential confounding factors.

References

1.Hearing loss prevalence and years lived with disability, 1990-2019: findings from the Global Burden of Disease Study 2019. Lancet. 2021;397(10278):996-1009. doi: 10.1016/s0140-6736(21)00516-x. PubMed PMID: 33714390; PubMed Central PMCID: PMCPMC7960691.

2.Wei X. Dietary magnesium and calcium intake is associated with lower risk of hearing loss in older adults: A cross-sectional study of NHANES. Front Nutr. 2023;10:1101764. Epub 20230314. doi: 10.3389/fnut.2023.1101764. PubMed PMID: 36998904; PubMed Central PMCID: PMCPMC10043168.

3.Agoritsas T, Merglen A, Shah ND, O'Donnell M, Guyatt GH. Adjusted Analyses in Studies Addressing Therapy and Harm: Users' Guides to the Medical Literature. Jama. 2017;317(7):748-59. doi: 10.1001/jama.2016.20029. PubMed PMID: 28241362.

4.Kernan WN, Viscoli CM, Brass LM, Broderick JP, Brott T, Feldmann E, et al. Phenylpropanolamine and the risk of hemorrhagic stroke. N Engl J Med. 2000;343(25):1826-32. doi: 10.1056/nejm200012213432501. PubMed PMID: 11117973.

5.Jaddoe VW, de Jonge LL, Hofman A, Franco OH, Steegers EA, Gaillard R. First trimester fetal growth restriction and cardiovascular risk factors in school age children: population based cohort study. Bmj. 2014;348:g14. Epub 20140123. doi: 10.1136/bmj.g14. PubMed PMID: 24458585; PubMed Central PMCID: PMCPMC3901421.

6.Liu H, Wang L, Chen C, Dong Z, Yu S. Association between Dietary Niacin Intake and Migraine among American Adults: National Health and Nutrition Examination Survey. Nutrients. 2022 Jul 25;14(15):3052. doi: 10.3390/nu14153052. PMID: 35893904; PMCID: PMC9330821.

7.Zhang Y, Lu J, Huang S, Chen Y, Fang Q, Cao Y. Sex differences in the association between serum α-Klotho and depression in middle-aged and elderly individuals: A cross-sectional study from NHANES 2007-2016. J Affect Disord. 2023 Sep 15;337:186-194. doi: 10.1016/j.jad.2023.05.073. Epub 2023 May 24. PMID: 37236270.

8.Mao Y, Li X, Zhu S, Geng Y. Association Between Dietary Fiber Intake and Risk of Depression in Patients With or Without Type 2 Diabetes. Front Neurosci. 2022 Jul 12;16:920845. doi: 10.3389/fnins.2022.920845. PMID: 36389250; PMCID: PMC9642095.

Methods:

Potential confounders were identified based on previous studies, clinical experience, p-values less than 0.05 in univariate analyses, and a change of more than 10% in the effect sizes of dietary niacin intake on hearing loss before and after adjusting for variables[5, 25-29],including age; sex; ethnicity/race; education level; marital status; household income; body mass index (BMI); smoking status; alcohol consumption status; ear infections; tinnitus; hearing protection; noise exposure; hypertension; stroke; diabetes; coronary heart disease; use of dietary supplements; dietary calorie intake; dietary protein intake;dietary carbohydrate intake;dietary fat intake;

Results :

The model was analyzed using multifactorial logistic regression, with all confounders accounted for in Model 3. The results indicated that dietary niacin intake was negatively associated with the risk of developing hearing loss.

Discussion:

The findings of this study indicated a negative correlation between dietary niacin intake and hearing loss. To ensure the reliability of the results, potential confounding variables were controlled for in Model 3 of our analyses. Additionally, the reliability and stability of the results were evaluated through multifactorial logistic regression analyses, stratification analyses, and sensitivity analyses.

Reviewer: 2

Dear Reviewer,

Thank you very much for your time involved in reviewing the manuscript and your positive comments and valuable suggestions to improve the quality of our manuscript. We have proofread the manuscript carefully, and rephrased the paragraphs and incorrect sentences. And we hope the revised manuscript could be acceptable for you.

COMMENTS FOR THE AUTHOR:

 Method Section:

1.It is still unclear how dietary niacin was obtained. Could you please provide more details?

Response 1）

We appreciate your valuable suggestions, which have enhanced the rigor of our research. Following your guidance, we have thoroughly revised the manuscript. For your reference, we have included detailed information on page 6-7, line106-119, highlighted in red for your review.

The data on dietary niacin intake were collected through 24-hour dietary recall interviews.NHANES dietary surveys were conducted by professionally trained personnel and were based on the Dietary Recall Interview measureme（https//www.cdc.gov/nchs/nhanes/guidemeasuring_guides_dri/measuringguides.htm).by converting to United States Department of Agriculture (USDA) standardized reference codes, the food intake aligned with the USDA Food and Nutrient Database for Dietary Studies (FNDDS) [22, 23]. The nutritional value of each individual's dietary intake was accurately calculated using the Computer-Assisted Dietary Interview (CADI) system and the Automated Multiple Pass Method (AMPM). The dietary information provided by the participants was assessed on two occasions by professionals in accordance with the Dietary Recall Interview Measurement Guide. The initial assessment was conducted in person at a mobile screening center, while the second assessment was conducted via telephone between three and ten days later. This approach to dietary assessment was extensively discussed at a workshop on NHANES data collection procedures and subsequently endorsed by experts in the field[24].

2.What is the goal of assessing meal frequency (line 113)? Additionally, the term "four group" in line 114 could be changed to "categorized into quartiles" for better clarity.

Response 2）

We would like to express our gratitude for your comprehensive and meticulous examination of our manuscript and the invaluable insights you have provided. We are grateful for the opportunity to elucidate and refine the manuscript in accordance with your recommendations. The objective of our study was to evaluate the relationship between dietary niacin intake and hearing loss (HL).Dietary niacin intake was assessed using a 24-hour dietary recall interview, initially conducted at a mobile examination center (MEC) and subsequently via telephone between three and ten days later. This method allowed for the evaluation of dietary intake data from two nonconsecutive days, leading to more accurate estimates of nutrient intake. Consequently, the dietary niacin intake data used in our study represented the mean of these two assessments.The term "meal frequency" was replaced with "dietary niacin intake" to more accurately reflect the objective of the study and to enhance the scientific rigour and clarity of the study. It is our hope that this change will be viewed favourably.Additionally, we greatly appreciate your valuable suggestion to change "the term 'four group' in line 114" to "categorized into quartiles" for better clarity. For a comprehensive overview, please refer to page 7, line 119-121. Thank you once again for your review and recommendations.

In this study, dietary niacin intake was averaged from two assessments. Participants were then categorized into quartiles based on their dietary niacin intake.

3.Background Section: 

The explanation of how niacin will be assessed among other vitamins/micronutrients is not yet clear. Please elaborate on the importance of niacin and why it is a focus of this study.

Response 3）

We are grateful for your comprehensive review of the manuscript and the constructive feedback you have provided. We have meticulously revised the manuscript in accordance with your suggestions and have highlighted them in red. For your reference, please direct your attention to page 3, line 25-32 of the manuscript for detailed information. We eagerly anticipate your favorable response.

Hearing loss poses a significant threat to human health, with its prevalence increasing annually. Niacin (vitamin B3) is an essential B vitamin that plays a crucial role in energy metabolism and cellular repair in the body. Additionally, it exerts a protective influence on the cells of the inner ear. A correlation between dietary niacin and hearing loss has been reported; however, the results remain controversial, necessitating further investigation. This study aimed to examine the potential association between dietary niacin intake and hearing loss in U.S. adults, providing a reference for dietary preventive management of hearing loss.

4.Method Section - Noise Exposure and Hearing Protection:

Could you provide more details on how noise exposure and hearing protection were determined?

Response 4）

We are grateful for your insightful suggestions and concur with your observations. We have undertaken a thorough review of the manuscript and made the requisite changes, which we have highlighted in red. Please refer to page 8, line 138-140for further details. We reiterate our appreciation for your invaluable guidance and advice.

Noise exposure is defined as exposure to loud noise within the past 24 hours.Hearing protection is classified according to the frequency of its use during noise exposure. The categories are: always, about half the time, seldom, and never.

5.Method and Discussion Sections - Types of Hearing Loss:

Could you briefly explain the differences between low-frequency hearing loss (LFHL), speech-frequency hearing loss (SFHL), and high-frequency hearing loss (HFHL) in both the method and discussion sections?

Response 5）

We are grateful for your perceptive recommendations and have incorporated the relevant information into the Methods and Discussion section as you suggested. We look forward to your feedback and anticipate a favorable response. We have implemente

---

## [Decision Letter · Decision Letter 1]

3 Dec 2024

PONE-D-24-05737R1L-shaped relationship between dietary niacin intake and hearing loss in US adults: National Health and Nutrition Examination SurveyPLOS ONE

Dear Dr. zhang,

Thank you for submitting your manuscript to PLOS ONE. After careful consideration, we feel that it has merit but does not fully meet PLOS ONE’s publication criteria as it currently stands. Therefore, we invite you to submit a revised version of the manuscript that addresses the points raised during the review process.

Dear Authors, 

There are language and other minor issues in the manuscript. You are strongly advised to revise and improve the manuscript language, fluency and grammar by professional English Editing Compony 

We look forward to receiving your revised manuscript.

Kind regards,

Aziz ur Rahman Muhammad

Academic Editor

PLOS ONE

Additional Editor Comments:

Dear Editor,

There are still language and other minor issue in the manuscript. You are strongly advised to revise and improve the manuscript language, fluency and grammar by professional English Editing Compony

Reviewers' comments:

Reviewer's Responses to Questions

**Comments to the Author**

1. If the authors have adequately addressed your comments raised in a previous round of review and you feel that this manuscript is now acceptable for publication, you may indicate that here to bypass the “Comments to the Author” section, enter your conflict of interest statement in the “Confidential to Editor” section, and submit your "Accept" recommendation.

Reviewer #1: All comments have been addressed

Reviewer #3: All comments have been addressed

2. Is the manuscript technically sound, and do the data support the conclusions?

Reviewer #1: Yes

Reviewer #3: Yes

3. Has the statistical analysis been performed appropriately and rigorously? 

Reviewer #1: Yes

Reviewer #3: Yes

4. Have the authors made all data underlying the findings in their manuscript fully available?

Reviewer #1: Yes

Reviewer #3: Yes

5. Is the manuscript presented in an intelligible fashion and written in standard English?

Reviewer #1: Yes

Reviewer #3: No

6. Review Comments to the Author

Reviewer #1: All my concerns raised during the initial review have been addressed and there are no dual publication, research ethics, or publication ethics issues.

Reviewer #3: There are multiple grammatical errors throughout the manuscript (spacing issues, punctuation, etc) that need to be corrected.

Line 67 does not make sense ("has been evidence to influence...").

Line 78 (hypothesis rather than assumption and need to fix the tense "we hypothesize that niacin would..."). This is the same issue with tense in line 79.

Line 91: please indicate if this was two cycles 2011-2012 and 2015-2016. It is difficult to determine if you are referring to data between 2011-2016. NHANES identifies this information as "cycles" and it should be referred to as such.

Line 146: appears information about how preexisting conditions were categorized is missing and needs to be updated.

Line 152: You have included information about total dietary calories, protein and carbohydates/fats. Were these also averaged similar to dietary niacin? This would need to be described in more detail. Also, you need to be specific about fat. Is this total fat or saturated fat? Also, how are these being represented. Grams or percentages of dietary intake?

7. PLOS authors have the option to publish the peer review history of their article (what does this mean? ). If published, this will include your full peer review and any attached files.

**Do you want your identity to be public for this peer review?** For information about this choice, including consent withdrawal, please see our Privacy Policy .

Reviewer #1: **Yes: ** Felix Gumaayiri Aabebe

Reviewer #3: No

---

## [Author Response · Author response to Decision Letter 2]

12 Jan 2025

Re:  PONE-D-24-05737

Dear Reviewers,

We sincerely appreciate your thorough review and insightful comments on our manuscript, titled “L-shaped Relationship Between Dietary Niacin Intake and Hearing Loss in US Adults: National Health and Nutrition Examination Survey” (ID: PONE-D-24-05737). In response to your feedback, we have made substantial revisions to strengthen the manuscript and are pleased to resubmit it for your consideration in PLOS ONE.

Enclosed, please find a detailed document addressing each of your comments. The revised manuscript includes the necessary changes, highlighted in red for your convenience. These revisions do not affect the statistical analyses or results.To address your concerns regarding grammatical errors and formatting, we engaged a professional English editing service staffed by native English speakers. They revised the manuscript to enhance its language, fluency, and clarity. Additionally, we thoroughly reviewed the revised text to ensure that all corrections were accurate and that the core research content remained intact.

We are grateful for your time and effort in reviewing our work. Should you require further clarification or additional information, we would be happy to provide it. We sincerely hope that our study will contribute meaningfully to the academic community.

We confirm that there are no conflicts of interest related to this work and that all authors have reviewed and approved the revised manuscript for submission.

Thank you once again for your valuable feedback

Best regards,

Yours sincerely,

Xue zhang

Email: zhangxue_jyfy@163.com

Mailing address: Affiliated Hospital of Jining Medical University, 89 Guhuai Road, Rencheng District, Jining, Shandong Province, China;

Reviewer: 1

Dear Prof. Felix Gumaayiri Aabebe

We deeply appreciate your thoughtful and thorough review of our manuscript. Your acknowledgment that all concerns raised during the initial review have been addressed and that there are no issues regarding dual publication, research ethics, or publication ethics means a great deal to us. Your constructive feedback has been instrumental in enhancing the quality and rigor of our work, and we are truly grateful for your time, expertise, and support throughout this process.

Reviewer: 2

Dear Reviewer,

We sincerely thank you for your detailed and insightful feedback on our manuscript. We deeply appreciate the time and effort you invested in providing such constructive comments. Your suggestions are invaluable, and we have addressed each of them comprehensively to enhance the manuscript's clarity, rigor, and accuracy. We hope the revised version meets your expectations and is suitable for publication.

COMMENTS FOR THE AUTHOR:

1.There are multiple grammatical errors throughout the manuscript (spacing issues, punctuation, etc) that need to be corrected.

Response 1）

We sincerely thank you for your constructive feedback regarding the grammatical errors and formatting issues in our manuscript. In response, we engaged a professional English editing service staffed by native speakers to revise the manuscript's language, fluency, and grammar, ensuring its accuracy and clarity. We carefully reviewed the revised text to confirm that no errors remain and that the core research content was preserved. We greatly appreciate your attention to detail and are confident that these revisions have strengthened the manuscript. If any further aspects require clarification or improvement, please do not hesitate to inform us. Thank you again for your valuable input.

2.Line 67 does not make sense ("has been evidence to influence...").

Response 2）

Thank you for your thorough and detailed review, as well as your valuable suggestions. We have revised this section to enhance clarity and coherence, ensuring the language accurately reflects the intended meaning and aligns with the supporting evidence. We appreciate your careful attention to this matter. The revision has been incorporated into the manuscript and highlighted in red for your convenience (see line 52, page 3). We look forward to your favorable consideration.

Modifying dietary intake has been shown to affect the risk of hearing impairment.

3.Line 78 (hypothesis rather than assumption and need to fix the tense "we hypothesize that niacin would..."). This is the same issue with tense in line 79.

Response 3）

Thank you for your valuable feedback regarding the phrasing and tense in lines 78 and 79. We have revised these sentences to improve clarity and ensure alignment with scientific conventions. We appreciate your detailed input, which has contributed significantly to enhancing the quality of our manuscript. The revisions have been incorporated into the manuscript and highlighted in red for your convenience (see lines 63–64 on page 4). We look forward to your favorable consideration.

We hypothesize that niacin intake is lower in individuals with HL. Additionally, the dose-response relationship between dietary niacin intake and hearing loss was assessed.

4.Line 91: please indicate if this was two cycles 2011-2012 and 2015-2016. It is difficult to determine if you are referring to data between 2011-2016. NHANES identifies this information as "cycles" and it should be referred to as such.

Response 4）

Thank you for your helpful comment regarding the clarification of the NHANES data cycles. We recognize the importance of using precise terminology and have revised the sentence to refer to the data as "cycles" in accordance with NHANES conventions. This revision ensures clarity about the specific NHANES cycles included in the study.

We appreciate your careful review and constructive suggestion. The revision has been incorporated into the manuscript and highlighted in red for your convenience (see lines 75–77 on page 5). We look forward to your favorable consideration.

NHANES data from the 2011–2012 and 2015–2016 cycles were used in the present cross-sectional study. Only hearing test data for adults aged 20–69 years were available for these two cycles

5.Line 146: appears information about how preexisting conditions were categorized is missing and needs to be updated.

Response 5）

Thank you for your insightful comment regarding the categorization of preexisting conditions. In response, we have revised the manuscript to provide greater clarity on how comorbidities were categorized. Specifically, we clarified that "comorbidities" include tinnitus, ear infection, hypertension, diabetes, stroke, and coronary heart disease. This revision ensures that the specific comorbidities accounted for in the analysis are clearly identified.We appreciate your careful review and constructive suggestion, which has enhanced the clarity of the manuscript. The revision has been incorporated into the manuscript and highlighted in red for your convenience (see lines 141–142 on pages 7–8). We look forward to your favorable consideration.

Model 2 was adjusted for sex, age, and comorbidities, including tinnitus, ear infections, hypertension, diabetes, stroke, and coronary heart disease.

6.Line 152: You have included information about total dietary calories, protein and carbohydates/fats. Were these also averaged similar to dietary niacin? This would need to be described in more detail.

Response 6）

Thank you for your comment regarding the averaging of total dietary calories, protein, carbohydrates, and fats, as well as your suggestion to further clarify this process. To ensure consistency with the assessment of dietary niacin intake, we have clarified that total dietary calories, protein, carbohydrates, and fats were averaged from two assessments, similar to dietary niacin. This revision enhances clarity and ensures a consistent methodological approach for these dietary variables.We appreciate your suggestion, which has improved the transparency of our methodology. The revision has been incorporated into the manuscript and highlighted in red for your convenience (see lines 128–131 on page 7 of the revised manuscript). We look forward to your favorable consideration.

Based on the dietary survey, 24-hour nutritional information was obtained for participants, including total dietary calories, protein, carbohydrates, and total fat. Similar to dietary niacin intake, these variables were averaged from two assessments to ensure consistency in the evaluation.

7.Also, you need to be specific about fat. Is this total fat or saturated fat?

Response 7）

Thank you for your comment regarding the clarification of fat intake. We have specified that the analysis refers to total fat intake, rather than saturated fat, and have made this distinction throughout the manuscript. The revised text now consistently refers to total fat intake. We appreciate your attention to this detail, which has enhanced the clarity of our study.

8.Also, how are these being represented. Grams or percentages of dietary intake?

Response 8）

Thank you for your insightful question regarding the representation of dietary intake. In response, we have clarified the units for the dietary variables in the manuscript. Specifically, dietary calorie intake is presented in kilocalories per day (kcal/d), while dietary protein, carbohydrate, and fat intakes are expressed in grams per day (g/d). Dietary niacin intake is presented in milligrams per day (mg/d). These units are now consistently specified throughout the manuscript, including in Table 1. We appreciate your attention to this detail, which has enhanced the clarity of our presentation.

---

## [Editor Report · Decision Letter 2]

3 Feb 2025

L-shaped relationship between dietary niacin intake and hearing loss in United States adults: national health and nutrition examination survey

PONE-D-24-05737R2

Dear Dr. zhang,

We’re pleased to inform you that your manuscript has been judged scientifically suitable for publication and will be formally accepted for publication once it meets all outstanding technical requirements.

Kind regards,

Aziz ur Rahman Muhammad

Academic Editor

PLOS ONE

Additional Editor Comments (optional):

Dear Authors

Thanks for revision
---

## [Editor Report · Acceptance letter]

PONE-D-24-05737R2

PLOS ONE

Dear Dr. zhang,

I'm pleased to inform you that your manuscript has been deemed suitable for publication in PLOS ONE. Congratulations! Your manuscript is now being handed over to our production team.

Kind regards,

on behalf of

Dr. Aziz ur Rahman Muhammad

Academic Editor

PLOS ONE